# Mononuclear Phagocytes, Cellular Immunity, and Nobel Prizes: A Historic Perspective

**DOI:** 10.3390/cells13161378

**Published:** 2024-08-19

**Authors:** Siamon Gordon, Annabell Roberti, Stefan H. E. Kaufmann

**Affiliations:** 1Department of Microbiology and Immunology, College of Medicine, Chang Gung University, Taoyuan 333, Taiwan; 2Sir William Dunn School of Pathology, University of Oxford, Oxford OX1 3RE, UK; annabell.roberti@path.ox.ac.uk; 3Max Planck Institute for Infection Biology, 10117 Berlin, Germany; kaufmann@mpiib-berlin.mpg.de; 4Max Planck Institute for Multidisciplinary Sciences, 37077 Göttingen, Germany; 5Hagler Institute for Advanced Study, Texas A&M University, College Station, TX 77843, USA; 6Charité—Universitätsmedizin Berlin, Corporate Member of Freie Universität Berlin and Humboldt-Universität zu Berlin, 10117 Berlin, Germany

**Keywords:** mononuclear phagocyte system, immunity, macrophages, dendritic cells, multinucleated giant cells, Nobel prizes, phagocytosis, plasma membrane receptors, homeostasis, history

## Abstract

The mononuclear phagocyte system includes monocytes, macrophages, some dendritic cells, and multinuclear giant cells. These cell populations display marked heterogeneity depending on their differentiation from embryonic and bone marrow hematopoietic progenitors, tissue location, and activation. They contribute to tissue homeostasis by interacting with local and systemic immune and non-immune cells through trophic, clearance, and cytocidal functions. During evolution, they contributed to the innate host defense before effector mechanisms of specific adaptive immunity emerged. Mouse macrophages appear at mid-gestation and are distributed throughout the embryo to facilitate organogenesis and clear cells undergoing programmed cell death. Yolk sac, AGM, and fetal liver-derived tissue-resident macrophages persist throughout postnatal and adult life, supplemented by bone marrow-derived blood monocytes, as required after injury and infection. Nobel awards to Elie Metchnikoff and Paul Ehrlich in 1908 drew attention to cellular phagocytic and humoral immunity, respectively. In 2011, prizes were awarded to Jules Hoffmann and Bruce Beutler for contributions to innate immunity and to Ralph Steinman for the discovery of dendritic cells and their role in antigen presentation to T lymphocytes. We trace milestones in the history of mononuclear phagocyte research from the perspective of Nobel awards bearing directly and indirectly on their role in cellular immunity.

## 1. Introduction

In keeping with the overall scope of the present review, we trace the growth of the subject during 50-year periods over the past 150 years in relation to selected Nobel awards, first awarded in 1901, in Physiology and Medicine, Chemistry and Physics (Figure 1A–D). Apart from the awards directly relevant to mononuclear phagocytes (MPs) in 1908 and 2011, we also note selected prizes in Immunology, Microbiology and Infectious diseases, as well as in Biochemistry/Metabolism, Genetics, Cell Biology, and methodologic advances which contributed substantially to their study (Figure 2).

During the mid-19th century, Charles Darwin laid the groundwork for his theory of the origin of species and for natural selection as the driver of evolution [1]. Before the first Nobel awards, Rudolph Virchow had already put forward a cell-based theory of health and disease [2,3], soon followed by the stunning achievements of Louis Pasteur in chemistry, microbiology, and vaccinations [4]. Also in the later 19th century, Claude Bernard became a major influence in physiology and experimental medicine through his concept of the stable “milieu interieur” [5], later termed Hom(e)ostasis by Walter Cannon [6]. These and other investigators influenced the thinking of Metchnikoff, a zoologist interested in embryology before his conversion to ”natural immunity” and comparative pathology [7]; Metchnikoff and Ehrlich shared Nobel awards in 1908 for their contributions to cellular and humoral immunity [8,9,10,11,12,13,14]. Other Nobel laureates in the early decades of the 20th century included the following (Table 1 and Figure 2):Emil von Behring, who received the first Nobel Prize for Physiology and Medicine in 1901 for introducing serum therapy as a passive vaccination against diphtheria and tetanus [15];Robert Koch, famed for his studies on tuberculosis and tuberculin-induced delayed hypersensitivity [16];Ramon Y. Cajal [17], who adapted Golgi’s silver staining method to map the intricate distribution of neurons in exquisite detail;Tuberculosis [18] and the non-neuronal microglia of the brain, identified by Pio del Hortega [19], have remained major topics of macrophage research to the present day.

Metchnikoff (1845–1916) can be justly considered to be the grandfather of macrophage research. His life has been documented in several readable biographies [14,20]. Born in the Ukraine, he worked in Russia on vaccines for anthrax before joining the Pasteur Institute in Paris [21]. His putative Eureka moment with marine invertebrates at Messina [22] consolidated his interest in macrophages and large phagocytic cells, both sessile and migratory, which responded to chemotactic stimuli generated by microbial invasion. By the use of microscopy and cell staining, he identified, among other observations, intracellular acid-fast Mycobacteria of unusual appearance within multinucleated macrophage giant cells in tuberculous granulomata, emphasizing host–pathogen interactions. Anticipating the later discovery of the microbiome [23], he postulated the intoxication of the host by deleterious products derived from intestinal bacteria [14] in contrast with beneficial flora such as Lactobacilli, which he advocated could promote human health in the form of yoghurt. Metchnikoff was prescient also in his fascination with aging, coining the term gerontology [24]. His remarkable insights into host–pathogen interactions extended to studies with his colleague Emile Roux on an experimental model of syphilis in primates [25]. He wrote several monographs on comparative pathology [7], expanding on lectures at the Pasteur Institute, and had a memorable meeting with Tolstoy, an ardent anti-vivisectionist, who had written a moving short story, *The death of Ivan Illich*, based on Metchnikoff’s brother [26]. His collaborators and successors at the Pasteur Institute [21] included Emile Roux [25], Alexandre Besredka [27], and Waldemar Haffkine [28]. Jules Bordet [29] subsequently received a Nobel award (Figure 1A) for the discovery of complement and Clemens von Pirquet [30] contributed to acute hypersensitivity allergic reactions; Charles Richet, a French physician who coined the term anaphylaxis, was awarded a Nobel Prize in 1913 [31].

**Figure 2 cells-13-01378-f002:**
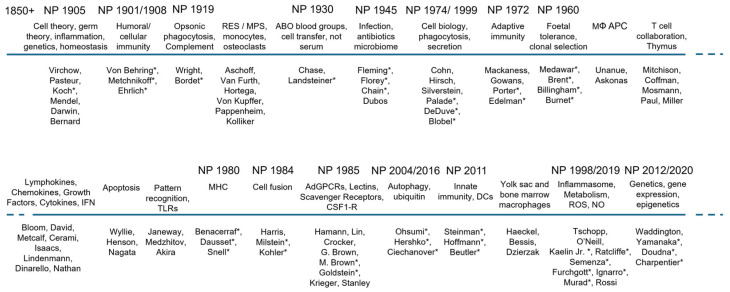
Selected milestones of mononuclear phagocyte research and cellular immunity. Note indirect contributions of numerous Nobel awards. Asterisks indicate Nobel Prize laureates. Abbreviations: RES: reticulo-endothelial system, MPS: mononuclear phagocyte system, AdGPCRs: adhesion G-protein coupled receptors [32,33,34,35,36,37,38,39,40,41,42,43,44,45,46,47,48,49,50,51,52].

In contrast with Metchnikoff’s citation for his Nobel award in cellular immunity, Paul Ehrlich (1854–1915) was cited for his work on humoral immunity [53] and his “side chain” cellular receptor theory of antibody induction [54]. He also used newly developed dyes [55] for differential staining of eosinophilic granulocytes, basophilic/mast cells, and polymorphonuclear neutrophilic leukocytes, initially called “microphages” by Metchnikoff. Finally, Ehrlich also performed seminal studies on the role of complement in humoral defense under the direction of antibodies. If the Nobel judges were hoping to reconcile the two opposing immunology camps through a joint award, they did not succeed at the time. Ironically, we now know that both sides were correct in that cellular and humoral immunity, as well as innate and acquired immunity, are closely interrelated in function. The cellular faction (Metchnikoff and his colleagues from the Pasteur Institute) favored the innate “unspecific” character, whereas the humoral faction (Emil von Behring, Paul Ehrlich, and colleagues) favored the humoral “specific” character as most critical. We now know that both the innate “unspecific”, as well as the acquired “specific” immune response are composed of both cellular and humoral components and that successful immunity depends on a highly regulated interplay between innate and adaptive immune responses.

## 2. The Reticulo-Endothelial System (RES)

In a comprehensive review in 1924, Ludwig Aschoff drew together the findings of many investigators who studied the intravital clearance of carmine particles by tissue macrophages in the liver (Kupffer cells), spleen, bone marrow, lymph nodes, and the lungs [56]; often forgotten later, an uptake by macrophages was also observed in endocrine organs, namely the adrenal and pituitary glands [56]. Walter Cannon [57] and Hans Selye [58] studied the importance of adrenaline in stress responses mediated by the sympathetic nervous system. Impressed by Metchnikoff’s work, Almroth Wright showed that agglutinins (antibodies) made particles and cells “tasty for eating”, coining the term opsonins/opsonization, enhancing phagocytosis in vivo and in vitro [59]. Much later, leukocyte receptors were shown to bind the Fc region of selected IgG antibodies, (originally defined by Rodney Porter and Gerald Edelman [60], who received Nobel awards for their structural studies); similar opsonic cell activation by distinct phagocyte complement receptors was observed for complement-derived C3bi, activated by IgG or IgM antibodies bound to cellular or microbial antigens [61], or via an alternative carbohydrate pathway [62]. IgE antibodies bound to allergens can induce histamine release from mast cells via FcR, triggering anaphylaxis [63]. Almroth Wright’s attempts to ”stimulate the phagocytes” to cure disease was spoofed by George Bernard Shaw in his play, *The Doctor’s Dilemma* [64]. We return to the characterization and functions of these opsonic leukocyte receptors below.

Following the belated appreciation of Gregor Mendel’s seminal discoveries on the genetic inheritance of discrete traits [65], Archibald Garrod coined the term “Inborn errors of metabolism” [66], revealing the great value of rare genetic diseases in deciphering pathophysiology in humans and other species. Later examples of discovery of dominant genes involved in myeloid phagocytes include chronic granulomatous disease [67], interferon receptor deficiencies [68], pyrinopathies [69], and lysosomal storage diseases [70]. An example of striking macrophage-related research is notable during this period; Peyton Rous, the discoverer of the Rous sarcoma virus, for which he received a Nobel Prize for many decades later, used the magnetic separation of Kupffer cells after the uptake of magnetic beads [71]. Florence Sabin, the first female medical graduate from Johns Hopkins and then member of the Rockefeller Institute, studied macrophage involvement in tuberculosis [72], as did Arthur Dannenberg, later at Johns Hopkins [73]. Also at Rockefeller was Nobel laureate Karl Landsteiner, discoverer of the ABO blood group antigen polymorphism [74]. In a landmark study, Merrill Chase and Landsteiner definitively showed that delayed hypersensitivity depended on the adoptive transfer of cells rather than passive immunization with serum antibodies [75].

At the same time, Howard Florey, Ernst Chain, and their colleagues at the Sir William Dunn School of Pathology in Oxford, followed up two earlier discoveries by Alexander Fleming of the bacteriolytic enzyme, lysozyme [76,77], and of an uncharacterized penicillium mold product with antibacterial activity [78]. This culminated in the clinically effective and safe antibiotic, penicillin, for which these three investigators shared a Nobel Prize [79]. Having, in his mind, solved the problem of bacterial infection, Florey, who edited an influential textbook on general pathology [80], turned, in the 1950s, to the host cellular response. He assigned his students, George Mackaness [81], James Gowans [82], and Henry Harris [83] the task of defining the role of macrophages, lymphocytes, and neutrophils, respectively. We shall pick up their subsequent contributions later. Another major figure in the history of immunological tolerance and organ transplantation, Peter Medawar, a zoology student at Oxford, built on his wartime studies of skin transplants in severely burnt patients in Glasgow, to investigate fetal tolerance to paternal antigens, culminating in a Nobel award with Leslie Brent and Rupert Billingham [84,85]; this was followed by a distinguished career, although handicapped by a premature stroke, as Director of the National Institute for Medical Research (NIMR) laboratory at Mill Hill in London. This became a major center for immunologic research over three quarters of a century [86] before moving recently to the Crick Institute. From a macrophage perspective, we single out the long-standing research interest at Mill Hill in influenza viral infection, the discovery of interferon by Alick Isaacs and Jean Lindenmann in the 1950s [87], and the groundbreaking studies by Philip D’Arcy Hart, who identified the role of mycobacterial inhibition of the phagosome fusion with lysosomes in the pathogenesis of tuberculosis [88]. John Humphrey and Deirdre Grennan made seminal contributions to the capture of capsulated bacteria, such as pneumococci, by spleen macrophages through polysaccharide recognition, important in T lymphocyte-independent initiation of immunity to infection [89]. Avrion Mitchison was a student of Medawar, working on transplantation rejection and later with George Snell and others on the use of inbred mouse strains, MHC expression, and T and B cell interactions [90]. Brigitte Askonas and Emil Unanue worked on antigen presentation, to be discussed below.

## 3. The Mononuclear Phagocyte System (MPS)

We return to Rockefeller in the 1960s, by now a university, in the runup to the establishment of the laboratory of the influential phagocyte research group by James G. Hirsch (neutrophils) [91], soon followed by Zanvil A. Cohn (monocytes and macrophages) [92], and later Ralph M. Steinman (dendritic cells) [93]. This myeloid leukocyte biology group emerged from the earlier recruitment to the Rockefeller Institute of the microbiologist René Dubos, whose benign presence persisted in the laboratory long after he turned his talents to ecology [94]. A French agrarian-trained admirer and later biographer of Pasteur, he joined Selman Waksman as a student at Rutgers University to investigate the interactions and antibacterial properties of soil organisms [94]. Waksman later won a Nobel award for streptomycin, which transformed the clinical treatment of tuberculosis [95]. A graduate student in the laboratory of Waksman, Albert Schatz, had first isolated streptomycin and both were jointly granted a patent for this drug. Waksman alone received the Nobel Prize in 1952 [96].

Dubos himself developed earlier antibiotics, gramicidin and tyrothricin, which unfortunately proved too toxic for clinical use [94]. After early spells at Rockefeller with Oswald Avery of DNA fame and at Harvard, Dubos returned to Rockefeller to study M. tuberculosis and Bacillus Calmette-Guérin (BCG) interactions with the host. He pioneered studies of the anaerobic flora of the gut and produced germ-free mice long before the microbiome became a general concern in immunity [23]. Dubos admired Metchnikoff, whose portrait took pride of place in his office and inspired all members of the phagocyte research group.

The range of research topics of this group has been documented in several books and reviews by Carol Moberg, secretary sequentially to Dubos [94], Hirsch [91], Cohn [92], and Steinman [93]. Hirsch was mainly interested in the study of neutrophil degranulation, phase contrast, electron microscopy of phagocytosis in a variety of species, and the bactericidal properties of histones [97]. His group made important contributions to morphologic studies of the intracellular infection of macrophages by a range of pathogens, including Legionnaire’s disease [98] and toxoplasmosis [99]. After early studies on the isolation of human monocytes [100], Cohn turned to the isolation and characterization of mouse peritoneal macrophages in cell culture. Both Hirsch and Cohn were influenced by the cell biological studies of Nobel laureates George Palade [101] and Christian de Duve [102] on the pancreatic secretory pathway and lysosomal digestion, respectively. The Cohn laboratory studied a range of cellular and immune functions, including phagocytosis, endocytosis, membrane traffic and recycling, cell fusion, lysosomal digestion and permeability; the group established that macrophages were not only “professional” phagocytes [103], but also potent secretory cells of enzymes, reactive oxygen and nitrogen species, and arachidonate metabolites [92]. Cohn’s interest turned to human macrophage infection by intracellular pathogens, such as tuberculosis, leprosy [104], Leishmaniasis [105], and HIV [106]. The Steinman–Cohn discovery of dendritic cells (DCs), culminating in the Nobel award to Steinman a few days after his death, will be dealt with below.

It is hard to overstate the international impact of this laboratory in macrophage research at the time, punctuated by a series of Leiden conferences organized by a Dutch collaborator, Ralph van Furth [107,108,109,110]. After a discussion at one of these meetings, it was decided that the term RES was no longer appropriate, and the family of cells was renamed the mononuclear phagocyte system [111].

### 3.1. Enter the Lymphocytes

Metchnikoff seems to have overlooked the lymphocyte in his observations and thought that after capture by phagocytes in organs such as spleen, the agglutinins, termed “antikörper” (antibodies) by Ehrlich, which were detected in blood after immunization, were also products of phagocytes [112]. James Gowans and his group at the Dunn School demonstrated the role of lymphatic recirculation and the delivery of smaller lymphocytes, rather than macrophages, to blood through the thoracic duct [82,113]. Florey, Gowans, and their students Alvin Volkman and Vincent Marchesi did, however, trace non-recirculating bone marrow-derived blood monocytes and macrophages directly to peripheral organs [114,115]. It seems remarkable that the thymus had no identified function until relatively late, when Jacques Miller published his first papers [116,117]. He also discovered T and B cell collaboration together with Jon Sprent [118]. The existence and properties of B (bursa/bone marrow), and thymus-derived lymphocytes and their heterogeneity (Th1, Th2, and even later, helper and cytotoxic, let alone memory, effector, and regulatory T lymphocyte subpopulations, as well as innate lymphoid cells) were only established after the 1960s. Their role in adaptive immunity dominated the awarding of Nobel prizes until 2011 (Table 1). It is interesting to compare these awards with those discoveries selected in published reviews of T [119] and B lymphocytes [120], as well as antibodies [121].

We emphasize here their essential collaboration with cells of the MPS in both the initiation and effector mechanisms of cell-mediated immunity. Australian immunologists have made major contributions in this field. We include Macfarlane Burnet, remembered for the clonal selection of antibody production, as well as self-/non-self-recognition, and virology [122,123], and Peter Doherty who, with the Swiss Rolf Zinkernagel, established the role of the major histocompatibility complex in cell-mediated immunity [124]. Donald Metcalf [125], also at the Walter and Eliza Hall Institute in Melbourne, and later, Richard Stanley [126], Australia-born, at the Albert Einstein College of Medicine in New York, played a major role in identifying the lineage-determining growth factors (CSF-1/M-CSF, and GM-CSF) for macrophage and granulocyte differentiation. George Mackaness and Robert North, both Australians at the Trudeau Institute in upstate New York, performed pioneering studies with BCG and Listeria monocytogenes infection models in genetically defined mouse strains [127]. They established that cell-mediated immunity to infection was antigen-dependent during induction, but nonspecific during expression, involving both T lymphocyte priming and the activation of macrophages. The soluble lymphokine factors, initially demonstrated independently by Barry Bloom and John David [128], and others, and their receptors were identified subsequently as appropriate methods became available. Charles Dinarello played an important role in characterizing interleukins/cytokines [129]. Mosmann and Coffman discriminated between Th1 and Th2 lymphocyte cytokines [130]. Carl Nathan played an important role in characterizing interferon gamma, the cytokine responsible for classical macrophage activation, see ref. [131] described further below, and Gray and Goeddel cloned its receptor [132].

### 3.2. Enter Dendritic Cells (DCs)

Prior to the discovery by Ralph Steinman and Zanvil Cohn that DCs are specialized, potent cells for antigen capture, processing, and presentation (APC) to naive B and T lymphocytes [93,133,134], it was assumed that the more abundant macrophages in adherent mononuclear phagocyte populations were responsible for the initiation of cell-mediated immunity. Emil Unanue, in particular, had demonstrated in important studies with Brigitte Askonas at Mill Hill [135] that a fraction of protein antigens was not completely degraded after the uptake by “macrophage” populations, as found in studies of endocytosis by Steinman, Silverstein, and Cohn [136]. It was only later that Alain Townsend, Pamela Bjorkman, and others identified the role of peptide binding by MHC molecules that protected antigenic epitopes for the presentation to CD4/CD8 T cells [137,138]. In their classic paper [139], Steinman and Cohn distinguished the rare DCs from conventional macrophages in mouse splenic digests, by phenotypic analysis and later demonstrated “immature” and “mature” functional states [93]. A subsequent paper with Michel Nussenzweig confirmed human DCs were also able to activate lymphocytes in allogeneic mixed lymphocyte cultures [140]. It took a decade before the unique ability of DCs to activate naïve lymphocytes became widely accepted [141]. Cultures of mouse bone marrow in GM-CSF and Interleukin 4, but not CSF1, gave rise to DC in vitro [142]. The hematopoietic origin of tissue and blood DCs has revealed considerable heterogeneity in DC subpopulations [143]. Studies of the “immune synapse” by which DCs induce B and T cell activation by MHC–peptide complexes, and effector recognition of target antigens by CD8 cytotoxic T cell receptors, continue to the present day [144]. DCs play a critical role in the decision to induce tolerance versus immune activation [145,146]. Artificial intelligence is being employed to identify tumor-specific peptides for personalized immunotherapy [147].

### 3.3. Enter Monoclonal Antibodies

The chemical and genetic elucidation of antibody structure and function was justly recognized by a series of Nobel awards in the 1960s and 1970s (Figure 1). Notably, Susumu Tonegawa was awarded the Nobel Prize in 1987 for his work at the Basel Institute for Immunology, in which he identified genetic rearrangement as the basis for antibody diversity. At Rockefeller, Henry Kunkel characterized the properties of monoclonal immunoglobulins in sera and urine derived from patients with multiple myeloma [148]. The roles of MP and DC in induction and effector mechanisms of both humoral and cellular immunity are intimately associated with the progress in research methods and our understanding of disease. Perhaps one of the most arresting advances in experimental and clinical immunology came from the application of somatic cell fusion to generate immortal hybrid cell lines producing monoclonal antibodies as tools to analyze phenotypic heterogeneity by flow and histochemical analysis. Henry Harris and John Watkins, in Oxford, started the ball rolling with their use of UV-irradiated Sendai viruses for the artificial fusion of different species of somatic cells to form homokaryons and heterokaryons, which could undergo subsequent unlimited proliferation as genetically labile and interspecific hybrids [149]. It soon became evident that differentiated cells extinguished their distinctive cell-specific properties unless fused with cells of a related lineage. Harris and Klein exploited cell fusion technology to demonstrate that malignancy was a recessive phenotypic trait and that fusion with normal diploid cells, such as fibroblasts, could transiently suppress this trait [150]. However, the tumor suppressor genes were readily lost in hybridomas because of chromosomal instability and the selection for growth in vivo or in vitro. Kohler and Milstein fused primary antibody-producing B lymphocytes with immortal myeloma cell lines in vitro to identify and isolate hybridoma clones by direct screening for monoclonal antibodies [151]. This method was optimized using polyethyleneglycol (PEG) instead of the Sendai virus and could be adapted to screen in advance for a desired functional phenotype. In the MPS field, it opened up the analysis of tissue heterogeneity in situ and provided biomarker antigens [152] for cell isolation by FACS and other methods of enrichment, leading to the discovery of novel plasma membrane receptor functions. An early mAb, Mac-1, produced by Timothy Springer [153,154] and directed at the CR3 receptor, led to the discovery of beta-2 integrins that are important in myeloid phagocyte recruitment and adhesion during inflammation. Monoclonal antibodies have also opened up therapeutic opportunities to manipulate the phenotype of immune cells, including mononuclear phagocytes [155]. In addition, somatic cell fusion is a natural property of mononuclear phagocytes in vivo, physiologically, in the generation of osteoclasts, and in immune and non-immune multinucleated giant cell (MNGC) formation, discussed further below.

### 3.4. Resident Tissue Macrophages

The F4/80 mouse antigen (Emr1), isolated by Austyn and Gordon 40 years ago [156], has become a widely used plasma membrane marker to identify macrophages in many mouse organs, from the embryo throughout adult life, normally and in a range of disease models [157,158]. It became the founding member of a family of adhesion G protein 7-transmembrane receptors [159]. The related human antigen (EMR2), which is absent in mice, has been implicated as a mechanoreceptor which undergoes a novel method of autoproteolytic activation [160]. Collaborative gene knockout studies with Hsi Hsien Lin and Joan Stein Streilein showed that the F4/80 molecule played a non-redundant role in peripheral tolerance in an anterior chamber-associated immune deviation (ACAID) model of delayed type hypersensitivity in the eye [161], extended to allografts and tumor implantation. Figure 3 and Figure 4 illustrate its expression in selected mouse tissues. With this groundwork, we repeated the method to identify and characterize a range of macrophage plasma membrane lectin-like (CD206, Siglec-1, Dectin1) and Scavenger receptors (SRA, MARCO and CD36) involved in adhesion, phagocytosis, endocytosis, pathogen entry, macrophage fusion, and cellular interactions [162]. By using a panel of surface markers, we were able to demonstrate the heterogeneous phenotypes of resident macrophages in a range of different organs (Figure 3).

**Figure 3 cells-13-01378-f003:**
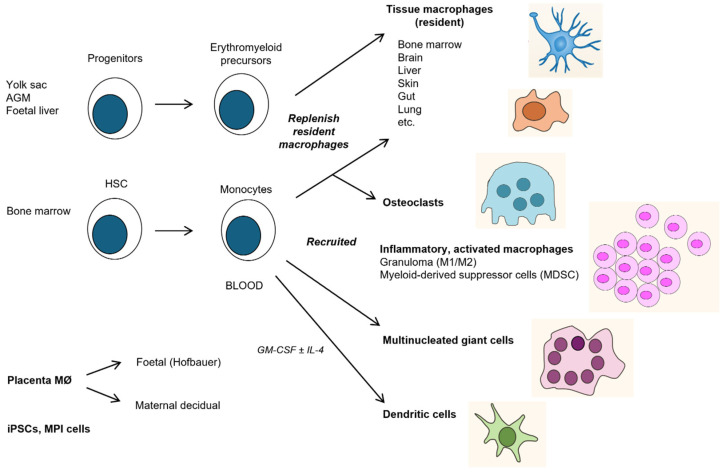
The mononuclear phagocyte system. The differentiation and distribution of monocytes, macrophages, DCs, and multinucleated giant cells (MNGCs) in vivo and in vitro, based mainly on F4/80 antigen expression in mice [157], on human placenta [163], and induced pluripotent stem cells (iPSC) [164]. Fejer has developed primary macrophage cell lines from mouse fetal livers, which are superior to tumor-derived cell lines (MPI cells) [165]. Abbreviations: AGM = aorta-gonad-mesonephros, HSC = hematopoietic stem cell, MDSC = myeloid-derived suppressor cell, GM-CSF = granulocyte-macrophage colony stimulating factor, MPI = Max Planck Institute.

Subsequent studies of temporal and spatial mRNA and protein gene expression in mouse and human tissue have considerably extended the concept that resident macrophages display distinct phenotypes within different tissue micro-environments [166,167]. The mechanisms which underlie their diversity include local interactions with neighboring differentiated cell types and the extracellular matrix, exposure to endogenous systemic stimuli, including the microbiome, which, in sum, regulates the epigenetic adaptation of macrophage gene expression [168,169].

**Figure 4 cells-13-01378-f004:**
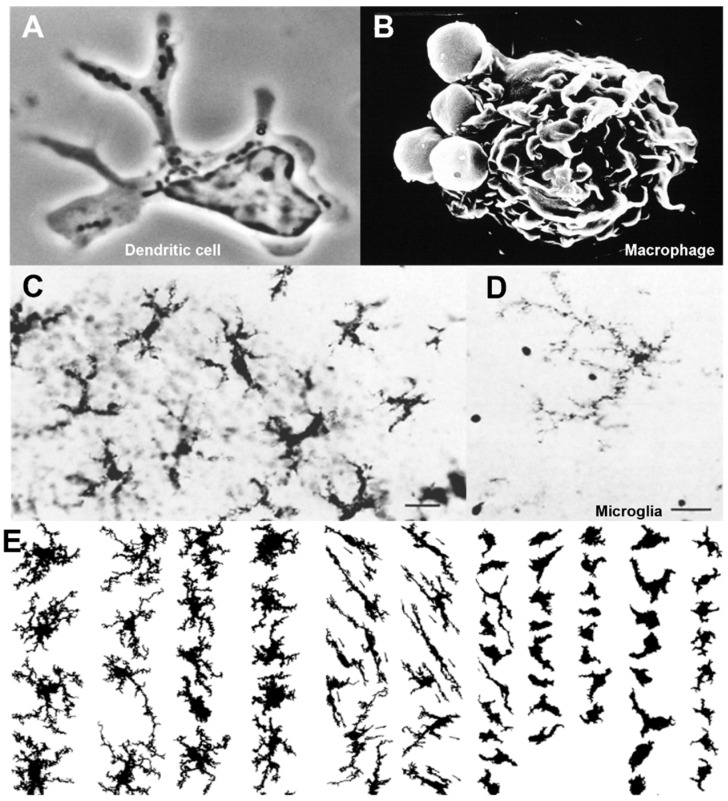
Morphology of mononuclear phagocytes. (**A**) Dendritic cell isolated from mouse spleen. Phase contrast microscopy reveals characteristic nuclear structure, cytoplasmic organelles, and dendritic processes [139]. (**B**) Mouse peritoneal macrophage-engulfing-antibody-coated sheep erythrocytes, note plasma membrane ruffling revealed by scanning electron microscopy. (**C**) F4/80+ Langerhans cells in the mouse epidermis [170]. (**D**,**E**) Mouse F4/80+ microglia in situ and montage. Note ramified processes of individual cells. (**E**) Striking regional morphologic heterogeneity in grey and white matter [171,172].

### 3.5. Development and Distribution

Studies of hematopoietic origins and lineage tracing in mice have given rise to a major paradigm shift in our understanding of the development of the MPS [173]. Progenitors of F4/80+ macrophages arise sequentially in the yolk sac (d8), AGM (d10.5), fetal liver (d10–12), and bone marrow before birth (d18–22), with widespread distribution throughout developing organs such as the brain [166,171]. An early origin from aortic endothelial stem cells and other mesenchymal cells has been established [174], and colony-stimulating factor1 (CSF-1)-responsive precursors of F4/80+ macrophages can be detected as early as d4 (SG, unpublished observations). Yolk sac-derived tissue-resident macrophages persist in many tissues such as the brain and epidermis throughout adult life, turning over locally and depending on different transcription factors such as Myb from those deriving from bone marrow [175,176], which populates organs with a high turnover, such as the gut. After birth, bone marrow-derived blood monocytes are the main source of recruitment on demand such as inflammation, infection, tissue injury, metabolic needs, and malignancy. Macrophage populations in adult organs are therefore chimaeras of varying embryonic and bone marrow origin. In several species, including humans, the CSF1R provides a pan monocytic lineage-determining surface marker which can be used to isolate subpopulations of monocytes and some DCs for further study, in conjunction with CD14 and CD16 antigens [177,178]. The mechanisms of circulation, tissue distribution, and migration in the fetus are not clear. After birth, bone marrow-derived monocytes use distinct adhesion molecules such as Beta2 integrins [179], L selectin, and CD31 during constitutive and induced extravasation and re-entry [180]. Monocyte/macrophage reserves can be mobilized from the spleen into blood and, during inflammation. from the peritoneal cavity to draining lymph nodes or the injured liver [181,182]. The niche for adhesion and repopulation may involve ligands induced on the endothelium, the extracellular matrix, or other cell types, e.g., hepatic stellate cells or the epithelium [183]. Blood monocytes can be delivered directly to the brain from bone marrow in the skull or via a leaky blood–brain barrier [184]. Monocytes recruited from blood contribute to the formation of osteoclast giant cells in bone and to MNGC in granuloma formation during infection [185]. Unlike macrophages, DCs recirculate from lymphatic circulation via the thoracic duct [186]. Recent studies have demonstrated that during stress such as fasting, blood monocytes can return to bone marrow for subsequent release into the blood upon refeeding [180].

### 3.6. Distinct Properties of Elicited Monocyte-Derived Mononuclear Phagocytes

Depending on the nature of acute or chronic stimuli, metabolic, inflammatory, infectious, immune, or malignant, circulating monocytes and newly recruited tissue macrophages display a range of altered phenotypes, reflecting their particular tissue environment. For example, sterile crystalline materials that are poorly degradable accumulate in lysosomes and stimulate foreign body giant cell formation [187]; bacteria, viruses, and helminths induce distinct cytokine responses and the formation of Th1 or Th2 types of granulomata; microbial products enhance a respiratory burst; the uptake of apoptotic or necrotic cells inhibit or enhance inflammation, respectively; and immune complexes and cytokines can enhance or suppress DC, T, and B cell activities. The concomitant responses of activated granulocytes, NK, innate lymphoid cells, DCs, CD4, and CD8 T cells can exacerbate and may exceed homeostatic limits, contributing to altered monocyte/macrophage proliferation, viability, clearance, secretory, cytotoxic, and repair functions. Below, we draw attention to some of the complex cellular aspects of these induced mononuclear phagocyte responses.

### 3.7. Re-Enter Complement

After earlier Nobel awards for complement and antibodies, research on humoral immunity centered almost exclusively on the characterization of plasma proteins. The later discovery of an “Alternative Complement pathway” by Pillemer, Müller-Eberhard, and Götze [188,189] and lectin-like recognition actually predated “classical” antibody-dependent activation in evolution. Progress in defining opsonic Fc and complement receptors on myeloid leukocytes provided an important link to cellular immunity, both innate and adaptive. Proteolytic cascades of complement, coagulation, and kinin pathways regulated phagocyte migration, cell activation, and target cell cytolysis via plasma membrane receptors and peptides. Analogous plasma proteins, such as Mannose-binding lectin (MBL) [190], calreticulin [191], and pentraxins [192], also serve as humoral pattern recognition receptors. The major source of many of these plasma proteins is the liver hepatocyte, but Harvey Colten and colleagues showed that macrophages were able to produce small amounts of all complement proteins during development, compatible with functions in local tissue environments [193]. Selected resident macrophages, for example, microglia in the CNS, express C1Q, which contributes to the sculpting of synapses by poorly understood mechanisms [194]. Recent discoveries, reviewed by Wright and Kemper, have shown that complement also plays an unexpected role intracellularly in innate and adaptive autoimmunity [195]. These involve T lymphocytes, as well as mononuclear phagocytes and DCs, and play a role in toll-like, TNF-related, and mitochondrial antiviral-signaling protein (MAVS) recognition and local responses. This discovery has rekindled interest in the role of locally produced complement as an integral part of cellular interactions during inflammasome activation, immunity and infection, as well as in metabolic homeostasis, development, and malignancy.

## 4. Cellular Functions of Mononuclear Phagocytes

### 4.1. Recognition, Uptake, and Degradation

Although many plasma membrane receptors and intracellular sensing mechanisms have been identified (Figure 5), the ability of MPs to discriminate a healthy from altered self at the cell surface or within intracellular membrane-bound compartments and the cytosol, remain poorly defined.

These myeloid cells are more promiscuous than lymphoid cells and less dependent on MHC-restricted selection of foreign or auto-antigens. Although the chemical composition of the walls of micro-organisms can be detected by a broad range of proteins, carbohydrates, lipids, and nucleic acid receptors, quality control of biosynthetic and intracellular transport products is less defined. We tend to fall back on abstractions such as “danger” [196] or “pattern recognition receptors” [197]; what can be stated with confidence is that there are few holes in the MP repertoire. The efficient clearance of particulates and apoptotic and necrotic cells by “professional” phagocytes is legendary [198], and “don’t eat me” signals such as signal-regulatory protein α (Sirpα)-CD47 interactions have stimulated the immunotherapeutic interest of oncologists [199]. Apart from the elegant early studies of Silverstein [200] and the ultrastructural studies by Grinstein and their colleagues [201], we still have little insight into molecular aspects of the phagocytic synapse [202,203], with or without opsonization, and the regulation of cargo internalization. Antibody-dependent enhancement of inflammation or infection via FcR and CR are important aspects of viral pathogenesis [204,205,206,207]. What is also intriguing is the ability of microglia to sculpt neuronal synapses and dendrites [208], and of stromal macrophages in bone marrow to ingest erythroid cell nuclei without compromising target cell viability [209]. Hidalgo and colleagues have demonstrated the uptake of effete mitochondria, by a heterophagic process, analogous to autophagy (Figure 6) [210]. Remarkably, intact mitochondria can be interchanged between adipocytes and macrophages without the loss of integrity or function [211]. Various names have been coined to stress variants of phagocytosis such as efferocytosis, pyroptosis, phagoptosis, ferroptosis, neuronophagy, and mitophagy. Apart from phago- and endolysosomal fusion, acidification, digestion, and membrane recycling during the intracellular invasion of macrophages are important aspects of pathogenesis [212] and determine immune cross-presentation and peptide loading of MHC molecules by DCs [213] and monocytes [214].

The biochemical mechanisms that control protein degradation by ubiquitination and delivery to proteosomes have previously attracted Nobel awards to Hershko and Ciechanover [218]; new methods have been developed to degrade selected cellular proteins for experimental and possible therapeutic purposes [219]. Finally, we stress the importance of endocytosis in lipid metabolism, recognized by Nobel awards to Brown and Goldstein, and the recycling of iron and other essential nutrients [220,221].

### 4.2. Biosynthesis, Gene Expression, Metabolism, Cell Activation, and Secretion

After terminal differentiation, MPs shut down DNA synthesis, but continue to express a wide range of mRNAs and proteins becoming highly active secretory cells [222] as well as phagocytes. Upon activation by innate and immune stimuli, they adapt to local and systemic environments to perform various homeostatic, inflammatory, and immune trophic and effector functions (Figure 7).

Tissue resident cells are long-lived, whereas elicited cells turn over more rapidly and are primed to generate more pro-inflammatory and cytotoxic radicals after further stimulation. Their secretory products include lysozyme [226], a range of neutral proteinases [227] to activate plasma cascades of complement, coagulation, and kinin generation, as well as arachidonate metabolites, leukotrienes, prostaglandins, and their derivatives. In addition, they produce chemokines, pro- and anti-inflammatory cytokines, and growth factors for lymphohematopoietic and endothelial cells [228], as well as protease inhibitors such as TIMP, alpha1 anti trypsin, and alpha2 macroglobulin. Activated macrophages also secrete collagenase and elastase, modify the extracellular matrix [229], and regulate fibroblast growth and repair. Their energy requirements of ATP are met by glycolysis, through the Krebs cycle and/or oxidative phosphorylation through mitochondrial respiration. The pioneering metabolic studies by Nobel laureates Otto Warburg [230], Hans Krebs, and Fritz Lipmann [231] are currently undergoing a renaissance in immunology [232].

Macrophage phenotype signatures are associated with a spectrum of polarized functional states [224,233]. Markers of innate stimulation include the enhanced expression of MARCO and CD200 [234]. Classical activation by interferon gamma, a product of CD4 Th1 lymphocytes and NK cells, induces a respiratory burst after local triggering, for example phagocytosis, to generate reactive oxygen and nitrogen radicals by activation of NADPH dehydrogenase [235] and iNOS [236], respectively. The vital roles of oxygen and nitrogen in metabolism have also been recognized by recent Nobel prizes.

An alternative activation pathway of macrophages [177,224,237,238] associated with allergy and nematode infestation and Th2 immunity, is mediated by IL-4/13 cytokines, which upregulate selected MHCII molecules, produce arginase and Chitinase-like proteins, and promote repair and fibrosis. Other macrophage functional states are associated with anti-inflammatory stimuli (IL10, TGF beta, corticosteroid, and prostaglandin E2) or immunoregulatory immune complexes. Tumor-associated macrophages (TAM) promote malignancy and metastasis [239,240,241]; temporal and spatial analyses of RNA and protein expression reveal considerable heterogeneity in TAM subtypes, relevant to potential immunotherapy [242].

Several hundred rare genetic disorders which result in familial auto-inflammatory and auto-immune syndromes have been discovered [69]. IL-1 beta release and pyroptosis are associated with periodic fevers and a range of clinical manifestations and can be treated with IL-1 receptor antagonist [243]. The discovery of inflammasome activation by the late Jurgen Tschopp [244], and of intracellular Nod-like, RIG-like, and DNA-sensing pathways, has spawned enormous activity in this field [245,246]. Life-threatening hyperinflammatory syndromes are associated with CD4 lymphocyte reconstitution during antiretroviral treatment of HIV/AIDS and severe pulmonary infections like SARS-CoV-2 [247,248]. It is worth noting that the term “cytokine storm” is a misnomer, since it encompasses many macrophage secretory products. Dexamethasone and several other repurposed drugs used for COVID-19 therapy act primarily on dysregulated macrophages, and the type 1 interferon pathway has been implicated in the genetic predisposition to critical outcomes of SARS-CoV-2 infection [68,249,250]. The potential role of MP in Long COVID sequelae should also be considered [251].

### 4.3. Cellular Immunity and Granuloma Formation

Mononuclear phagocytes play a prominent role in the formation and persistence of organized structures known as granulomas, which are not tumors, but assemblies of heterogeneous monocytes, macrophages, and DCs, together with other myeloid and lymphoid cells, fibroblasts, and blood vessels, embedded in an extracellular matrix [252,253,254]. Their morphology has been recognized since early descriptions by pathologists, and they cover a range of immunologic and inflammatory host responses to infection, foreign or host-derived non-degradable materials, and autoimmune disorders [255]. Although not malignant, per se, they have features in common with tumor-associated macrophages and can inflict considerable tissue injury upon their local environment, promote fibrosis, and even progress to true tumor formation, as in Schistosomiasis [256]. A characteristic feature is macrophage multinucleation, the result of cell fusion induced by Th1 and Th2 lymphocytes and their secretory products [257]. Metchnikoff recognized MNGC, known as Langhans giant cells, containing intracellular M. tuberculosis, with evident impairment of mycobacterial division [8,258]. There has been recent significant progress in studying the mechanism of fusion and its impact on the macrophage phenotype in a variety of cellular model systems, in vitro [259], and in autoimmune granulomatosis, in vivo [254]. This is a fascinating field for the further study of cellular biology and immunology, relevant to the impact of environmental pollutant particles on human health [260].

## 5. Discussion

Our review highlights the importance of evolution [261] and homeostasis [262] in considering the roles of macrophages and closely related DCs and MNGCs in pathophysiology and natural selection. We have traced the growth of knowledge from the emergence of these cells as a mononuclear phagocyte system, during development and throughout life, and described selected cellular and molecular properties. These widely distributed phagocytes recognize and interact with every other cell type of the body in their micro-environment, including microbes and foreign particulates, through plasma membrane contact, internalization, and secretion (Figure 8). Reciprocal interactions between the MPS and neuroendocrine [263], cardiovascular [264], and gastrointestinal systems [265], for example, extend the role of mononuclear phagocytes beyond host defense, inflammation, and immunity. Within the lymphohematopoietic system, it is important to recognize that macrophages are integral parts of all immunity, whether innate or adaptive, cellular or humoral, and participate in many disease processes as monocytes, tissue macrophages, DCs, or MNGCs. These “non-immune” diseases include metabolic processes such as atherosclerosis [266] and fatty liver disease [267], neurodegeneration such as Alzheimer’s disease [268], and malignancy [240]. We have stressed the value of genetic, cellular, and molecular analysis in deciphering and manipulating the MPS in situ and in vitro.

Although it is well-recognized that macrophage-like cells are ancient, long preceding the emergence of lymphocytes, clonal selection, and gene rearrangement, there has been much less investigation of myeloid cell evolution. Compelling studies by Max Cooper [269] and colleagues have brought insights into lymphocyte evolution; presumably, the role of MHC in peptide capture and antigen presentation by DCs should coincide with the evolution of lymphocyte activation. Comparative genomics and DNA analysis are making it possible to trace the origin and migration of human populations and the role of pandemics in evolution. The recent Nobel award to Paabo [270] and the studies of Quintana-Murci [271] and Barreiro [272] suggest that the MPS contributed significantly to selection by infectious disease. The hypothesis by Lynn Margulis [273,274] that mitochondria and chloroplasts are endosymbionts of captured bacteria is now widely accepted and the retention of retroviral sequences in the human genome [275] provides further evidence for the role of gene transfer and selection in evolution. Experiments to model the origin of eukaryotic cells [276] reveal cells which display macrophage-like features.

Apart from the specific Nobel prizes in the macrophage field in 1908 and 2011, we have shown how important many other awards in immunology and related subjects, including technological advances, have been in promoting and documenting the progress of research on MPs. This acknowledgement attests to the wider legacy of the Nobel awards in cellular, biomedical, and chemical research.

From the viewpoint of MPS contributions to cellular immunity, there has been considerable incremental progress in cell and molecular biology, but promising insights into biology and therapeutic applications need further development. Its relevance to the human lifespan and to host–pathogen and other environmental interactions extends beyond immunity and will be explored in detail elsewhere.

## 6. Conclusions

We have outlined some of the evidence that the MPS is a unique dispersed organ consisting of a family of closely related cells, interacting with every other system in the body, consistent with its primary role in physiological homeostasis, cellular immunity, and disease. Perturbation of its trophic cellular and molecular functions may contribute to the mechanisms of aging and malignancy and provide opportunities for therapy through genetic and immunologic manipulation. We have stressed the direct and indirect benefits of many Nobel awards to achievements in MPS research and noted some of the factors that promote scientific advances in this, as in other fields, where fundamental discoveries can lead to clinical applications. Our historic perspective has confirmed the international nature of scientific discovery and its dependence on institutional excellence, ready access to education, and freedom of movement and the exchange of ideas. Finally, we suggest that the evolutionary origins of macrophage diversification need more research to establish its rightful role in biology, beyond immunity.

## Figures and Tables

**Figure 1 cells-13-01378-f001:**
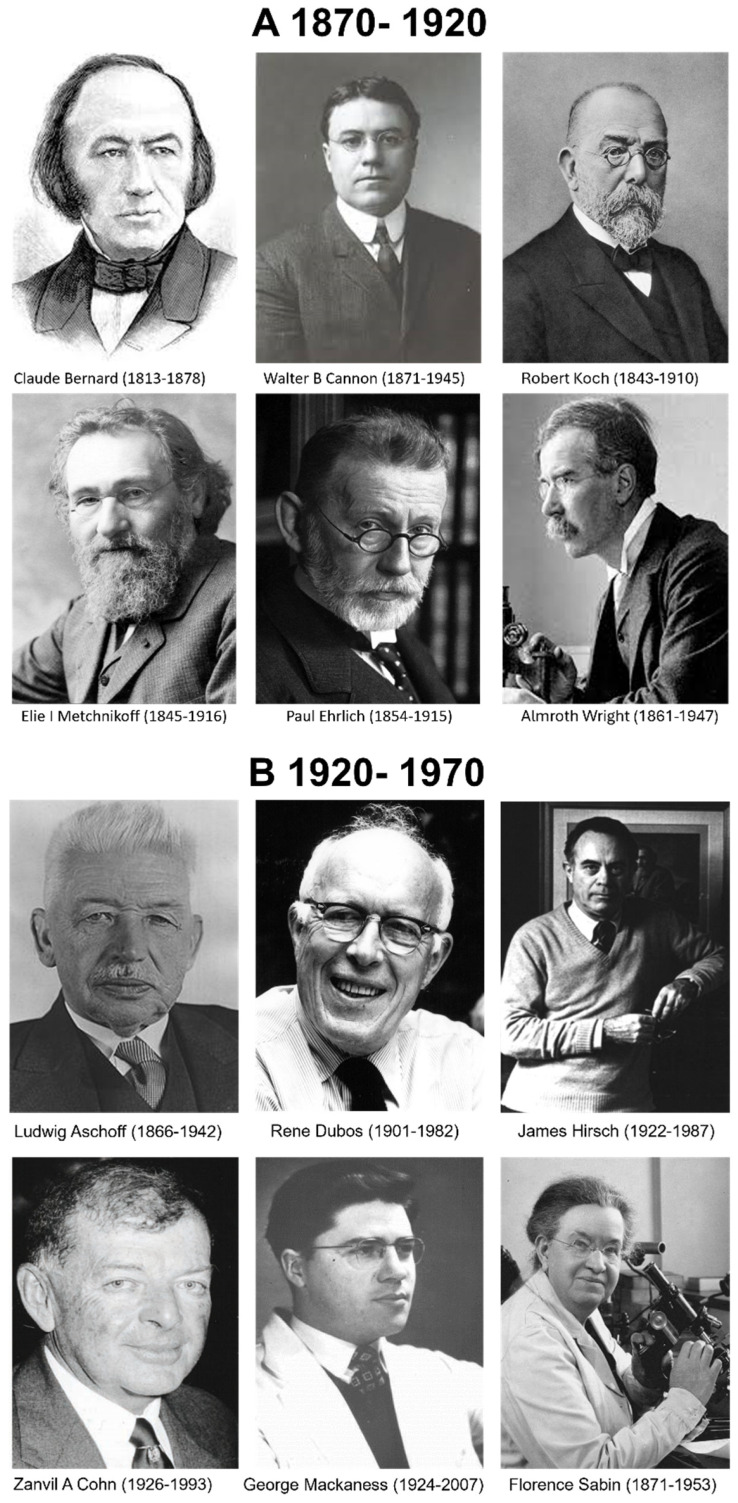
Selected historical figures who contributed to understanding the role of mononuclear phagocytes in cellular immunity. (**A**) Years 1870–1920, (**B**,**C**) 1920–1970, and (**D**) 1970–2020. See text and attachments for details. In this manuscript, we use the French spelling Elie Metchnikoff instead of the Russian version Ilya Metchnikov.

**Figure 5 cells-13-01378-f005:**
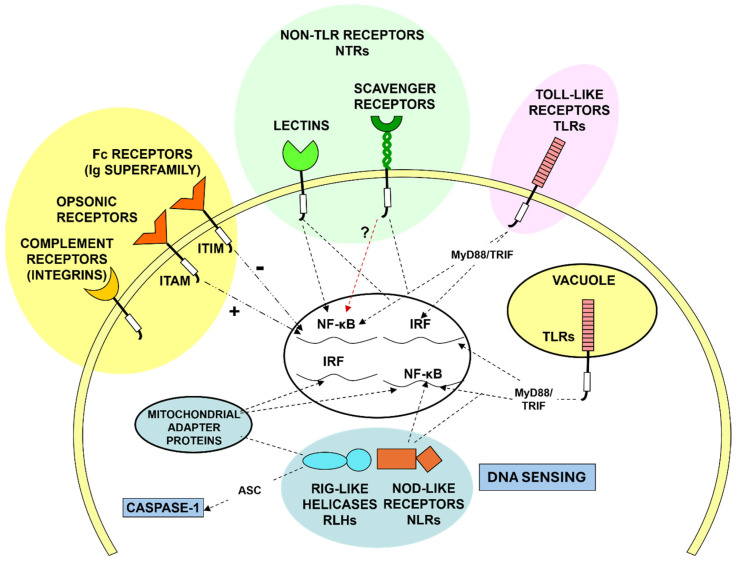
Sensing environment and mononuclear phagocyte responses. Schematic representation of plasma membrane opsonic Fc and complement receptors, Toll-like receptors (TLRs), and non-opsonic lectin-like and Scavenger receptors. Signaling pathways of opsonic receptors depend on ITAM and ITIM cytoplasmic domains for activatory and inhibitory responses. Cytoplasmic sensors include mitochondrial antiviral signaling (MAVS) proteins, RIG-like helicases, NOD-like receptors, and components of the DNA sensing CGAS pathway. Inflammasome activation depends on ASC and Caspase-1. Abbreviations: ITAM = immunoreceptor tyrosine-based activation, ITIM = immunoreceptor tyrosine-based inhibitory motif, NF-κB = nuclear factor kappa-light-chain enhancer of activated B cells, IRF = interferon regulatory factors, RIG = retinoic acid-inducible gene/protein; NOD = nucleotide binding and oligomerization domain; Adapted from A Pluddemann.

**Figure 6 cells-13-01378-f006:**
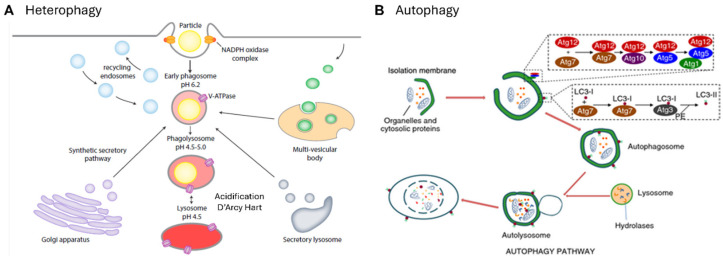
Heterophagy and autophagy: (**A**) Heterophagy is a defining property of mononuclear phagocytes. The schema shows key stages in the process of recognition, membrane, and vesicular fusion and fission, and their recycling during cargo internalization and product secretion. Two important outcomes of phagocytosis are acidification/digestion and the generation of oxygen and nitrogen radicals in host defense, critical determinants of intracellular infection. (**B**) The autophagy pathway employs analogous mechanisms to recognize, isolate, and degrade damaged intracellular constituents. The genetic and molecular aspects of autophagy have attracted considerable attention recently [215]. Adapted from [215,216,217]. Abbreviations: Atg = autophagy-related gene, LC3 = microtubule-associated proteins 1A/1B-light chain 3.

**Figure 7 cells-13-01378-f007:**
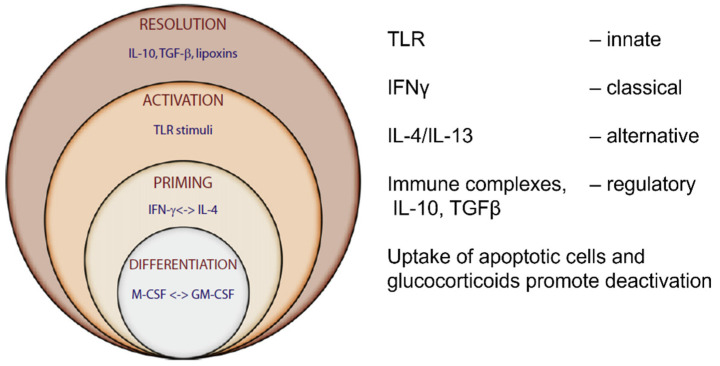
Paradigm of macrophage polarization. Macrophage gene expression is regulated differentially after priming by the Th1/2 cytokines and interferon gamma versus IL4/13, followed by a local innate phagocytic stimulus. The spectrum of activation extends from a signature cluster of genes characteristic of classical or alternative activation to deactivation by IL-10, tgf beta, glucocorticoid steroids [177], or by the uptake of apoptotic cells [46,223]. Polarization of macrophages mirrors that of T lymphocytes [224]. Adapted from [225].

**Figure 8 cells-13-01378-f008:**
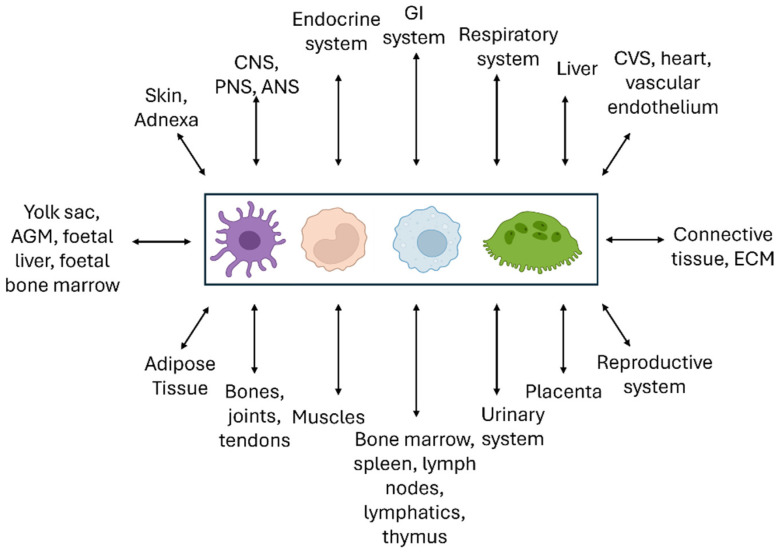
Interactions of the MPS with other systems. We place the cells of the MPS at the center of the homeostatic networks which interact reciprocally with all other cellular systems of the body. Illustrative examples of such systemic tissue interactions are given in the text.

**Table 1 cells-13-01378-t001:** Selection of Nobel awards in Physiology or Medicine and Chemistry and Physics which bear indirectly or directly on mononuclear phagocytes.

Year	Names	Topic	Comment
1901	Von Behring	Diphtheria	Antibodies
1905	Koch	TB	Cell-mediated
1908	Metchnikoff, Ehrlich	Phagocytosis, Receptors	Cell-mediated and antibodies
1912	Carrel	Work on vascular suture and the transplantation of blood vessels and organs	Cell culture
1913	Richet	Anaphylaxis	Cell-mediated and antibodies
1919	Bordet	Complement	Humoral
1930	Landsteiner	Blood groups	Antibodies
1931	Warburg	Respiration	Metabolism
1945	Fleming, Chain, Florey	Penicillin	Antibiotics
1950	Kendall, Reichstein, Hench	Corticosteroids	Anti-inflammatory
1951	Theiler	Yellow fever vaccine	Adaptive
1952	Waksman	Streptomycin	Antibiotic TB
1953	Krebs, Lipmann	Citric acid cycle, coenzyme A	Metabolism
1954	Enders, Weller, Robbins	Polio virus	Isolation
1960	Burnet, Medawar	Self and non-self, tolerance	Cell-mediated, transplantation
1966	Rous, Huggins	Viral malignancy	Magnetic beads macrophage isolation
1972	Edelman, Porter	Antibodies	Structure
1974	Claude, de Duve, Palade	Cell structure	Cell biology, transmission EM
1975	Baltimore, Dulbecco, Temin	Viral replication	NF-κB pathway
1976	Blumberg, Gajdusek	Hepatitis, Prions	Infection
1980	Benacerraf, Dausset, Snell	Histocompatibility antigens	Genetics
1982	Bergstrom, Samuelsson, Vane	Prostaglandins	Regulatory
1984	Jerne, Kohler, Milstein	Monoclonal antibodies	Specific targeting
1985	Brown, Goldstein	Cholesterol	Scavenger receptor
1987	Tonegawa	Genetic control of antibodies	Lymphocytes
1990	Murray, Thomas	Transplantation	Cell-mediated
1994	Gilman, Rodbell	G-proteins	Signalling
1996	Doherty, Zinkernagel	MHC peptide recognition	Cell-mediated and lymphocytes
1998	Furchgott, Ignarro, Murad	NO	Effector metabolite
1999	Blobel	Protein signal peptide	Membrane transport
2001	Hartwell, Hunt, Nurse	Regulators of cell cycle	Cell cycle
2002	Brenner, Horvitz, Sulston	Genetics of organ development, programmed cell death	Evolution, *C. elegans*
2005	Marshall, Warren	Helicobacter pylori infection	Peptic ulcers
2008	Zur Hausen, Barre-Sinoussi, Montagnier	HPV and HIV	CD4 depletion, opportunistic infection
2011	Hoffmann, Beutler, Steinman	Insect, TNF, TLR, DCs	Innate immunity and antigen presentation
2012	Gurdon, Yamanaka	Reprogramming differentiation	iPSC
2013	Rothman, Schekman, Sudhof	Vesicular trafficking	Cell biology
2016	Ohsumi	Autophagy	Macrophages
2018	Allison, Honjo	Cancer immunotherapy	Checkpoint inhibitors
2019	Ratcliffe, Kaelin, Semenza	Oxygen sensing	Respiratory burst
2020	Houghton, Alter, Rice	Hepatitis C virus	Virus
2021	Patapoutian, Julius	Receptors for temperature and touch	Receptor biology
2022	Paabo	Genomes of extinct hominins and human evolution	Evolution and genetics
Chemistry and Physics			
1902	Fischer	Sugar and purine synthesis	Biochemistry
1946	Stanley, Northrop	Tobacco mosaic virus crystallization	Viral structure
1948	Tiselius	Electrophoresis	Method
	Pauling	Protein bonds	Structural Biology/Chemistry
1958	Sanger	Proteins	Sequence
1962	Perutz, Kendrew	Protein Structure	Analytic
1972	Anfinsen	Ribonuclease	Protein strucure
1980	Berg, Gilbert, Sanger	DNA	Sequencing
1982	Klug	Crystallography	Protein structure
1993	Mullis, Smith	PCR and mutagenesis	Nucleic acid
2004	Ciechanover, Hershko, Rose	Protein ubiquitination	Degradation
2009	Ramakrishnan, Steitz, Yonath	Ribosome	Structural biology
2008	Shimomura, Chalfie, Tsien	Green fluorescent protein	Detection, microscopy
2012	Lefkowitz, Kobilka	GPCRs	Signalling
2018	Arnold, Smith, Winter	Enzymes, phage display of peptides and antibodies	Directed evolution
2020	Charpentier, Doudna	Genome editing	CRISPR
2022	Bertozzi, Meldal, Sharpless	Click chemistry and biorthogonal chemistry	Protein and therapeutics

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
