# Peer review of "Mononuclear Phagocytes, Cellular Immunity, and Nobel Prizes: A Historic Perspective"

_cells, 2024, doi:10.3390/cells13161378_

Round 1

Reviewer 1 Report

Comments and Suggestions for Authors

The authors have provided a detailed, interesting and accurate summary of key discoveries in the field of mononuclear phagocytes and cellular immunity, with an emphasis on the contributions of Nobel laureates. The authors are eminent and authoritative scientists in the field and overlook the topic very well. As far as I can say, most major discoveries were discussed and their presentation is objective and balanced. I enjoyed reading this fine review and learned much.

I have only a few remarks:

- Jacques Miller is not only famous for having elucidated the function of the thymus, he also discovered T B cell collaboration in 1972 together with Jon Sprent. The authors attribute this to Av Mitchison, who I think was a bit later. Also the authors mention in line 194 that Mitchison's work will be discussed below, but this is not the case.

- in line 454, the alternative complement pathway is explicitly mentioned. It would seem approproate to mention its discoveres, Pillemer, Mueller-Eberhard and Goetze

- The authors mention several discoveries related to viruses, which were important for immunology. They might consider mentioning Stanley and Northrop who receive the Nobel price in Chemistry 1946 for crystallizingTobacco mosaic virus

- Maybe the Chemistry Nobel price for GFP in 2008 to Shimomura, Chalfie and Tsien is worth mentioning, as it was critical for the study of phagocystosis

- Readers not only interested in the past but also in the future would appreciate some ideas on recent discoveries fundamental enough to warrant the "next" Nobel price. This will be highly subjective of course, but it is a chance to highlight that research is still ongoing and much is left to be discovered in this exciting field of research.

Author Response

We appreciate the comments and suggestions of both reviewers as shown below.

Referee 1 The authors have provided a detailed, interesting and accurate summary of key discoveries in the field of mononuclear phagocytes and cellular immunity, with an emphasis on the contributions of Nobel laureates. The authors are eminent and authoritative scientists in the field and overlook the topic very well. As far as I can say, most major discoveries were discussed and their presentation is objective and balanced. I enjoyed reading this fine review and learned much.

All re-worked sections are marked in red.

I have only a few remarks:

- Jacques Miller is not only famous for having elucidated the function of the thymus, he also discovered T B cell collaboration in 1972 together with Jon Sprent. The authors attribute this to Av Mitchison, who I think was a bit later. Also the authors mention in line 194 that Mitchison's work will be discussed below, but this is not the case.

This has been corrected in the text (Ref. 118 and 90).

- in line 454, the alternative complement pathway is explicitly mentioned. It would seem approproate to mention its discoveres, Pillemer, Mueller-Eberhard and Goetze

This has been addressed by adding references 188 and 189.

- The authors mention several discoveries related to viruses, which were important for immunology. They might consider mentioning Stanley and Northrop who receive the Nobel price in Chemistry 1946 for crystallizingTobacco mosaic virus

This has been added to table 1.

- Maybe the Chemistry Nobel price for GFP in 2008 to Shimomura, Chalfie and Tsien is worth mentioning, as it was critical for the study of phagocystosis.

This was already included in table 1.

- Readers not only interested in the past but also in the future would appreciate some ideas on recent discoveries fundamental enough to warrant the "next" Nobel price. This will be highly subjective of course, but it is a chance to highlight that research is still ongoing and much is left to be discovered in this exciting field of research.

A paragraph has been added right before the conclusion section.

Reviewer 2 Report

Comments and Suggestions for Authors

The review article explores a comprehensive historical perspective on the role of mononuclear phagocytes in cellular immunity, emphasizing significant milestones recognized by Nobel Prizes. It delves into:

1)    Historical Context: the development of the mononuclear phagocyte system (MPS) and its evolution over the past 150 years.

2)    Nobel Prizes and fundamental discoveries: detailed accounts of Nobel Prize-winning discoveries relevant to the MPS, such as the work of Elie Metchnikoff, Paul Ehrlich, Jules Hoffmann, Bruce Beutler, and Ralph Steinman.

3)    Cellular and molecular functions: in-depth discussions on the diverse roles of mononuclear phagocytes, including recognition, uptake, degradation, biosynthesis, gene expression, metabolism, and cellular activation.

4)    Interactions and homeostasis: the interplay between mononuclear phagocytes and other cellular systems in maintaining homeostasis and their involvement in various diseases.

Furthermore, the article includes eight figures and one table. The figures are very interesting and illustrate the main points highlighted in each section. A particular highlight is Figure 1, which shows selected historical figures who contributed to understanding the role of mononuclear phagocytes in cellular immunity. The article is well-referenced, indicating a thorough review of the literature. It includes a comprehensive list of 273 references, providing a solid foundation for the discussed topics. The references span from classic foundational studies to recent advancements, reflecting the depth and breadth of the research.

 Although the historical context is complete and informative, the article is long and somewhat exhaustive. Although valuable, the extensive detailing of historical landmarks can make the reading experience tiring for some readers. I suggest reducing the text referring to the historical sections to make the reading more fluid and engaging.

Minor concerns:

  - There is no Topic 3, which disrupts the logical flow of the sections.

  - Subtopics 3.4 and 3.5 have repeated titles, which can confuse.

Author Response

We appreciate the comments and suggestions of both reviewers as shown below. 

All re-worked sections are marked in red in the manuscript.

Referee2: Comments and Suggestions for Authors

The review article explores a comprehensive historical perspective on the role of mononuclear phagocytes in cellular immunity, emphasizing significant milestones recognized by Nobel Prizes. It delves into:

1)    Historical Context: the development of the mononuclear phagocyte system (MPS) and its evolution over the past 150 years.

2)    Nobel Prizes and fundamental discoveries: detailed accounts of Nobel Prize-winning discoveries relevant to the MPS, such as the work of Elie Metchnikoff, Paul Ehrlich, Jules Hoffmann, Bruce Beutler, and Ralph Steinman.

3)    Cellular and molecular functions: in-depth discussions on the diverse roles of mononuclear phagocytes, including recognition, uptake, degradation, biosynthesis, gene expression, metabolism, and cellular activation.

4)    Interactions and homeostasis: the interplay between mononuclear phagocytes and other cellular systems in maintaining homeostasis and their involvement in various diseases.

Furthermore, the article includes eight figures and one table. The figures are very interesting and illustrate the main points highlighted in each section. A particular highlight is Figure 1, which shows selected historical figures who contributed to understanding the role of mononuclear phagocytes in cellular immunity. The article is well-referenced, indicating a thorough review of the literature. It includes a comprehensive list of 273 references, providing a solid foundation for the discussed topics. The references span from classic foundational studies to recent advancements, reflecting the depth and breadth of the research.

 Although the historical context is complete and informative, the article is long and somewhat exhaustive. Although valuable, the extensive detailing of historical landmarks can make the reading experience tiring for some readers. I suggest reducing the text referring to the historical sections to make the reading more fluid and engaging.

We have discussed this suggestion among ourselves but prefer to leave the text as it is considering the historic emphasis of the article. We have used tables, figures and text to illustrate and highlight the evolution of the subject.

Minor concerns:

  - There is no Topic 3, which disrupts the logical flow of the sections.

 This has been corrected.

  - Subtopics 3.4 and 3.5 have repeated titles, which can confuse.

This has been corrected.